# Long-Term Leases vs. One-Off Purchases: Game Analysis on Battery Swapping Mode Considering Cascade Utilization and Power Structure

**Guohao Li \* and Tao Wang** 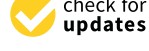

School of Management, Jiangsu University, Jingkou District, Zhenjiang 212013, China
*   Correspondence: guohaolee@ujs.edu.cn

**Abstract:** The electric vehicle industry faces intense competition and the sustainability problem. In order to obtain a differential competitive advantage, enterprises actively promote the battery swapping mode (BSM) to respond to cost pressures caused by the mismatch between demand and supply. Considering cascade utilization, the Stackelberg game models of electric vehicle supply chain under three different scenarios, in the secondary supply chain consisting of a battery manufacturer (BM) and a vehicle manufacturer (VM), were constructed, respectively. Additionally, then, through the contrastive analysis of differential power structures, the influence of power structures and related parameters on the optimal pricing strategy and enterprise profits of both parties in the supply chain were studied and compared. The conclusions show that the dominance of the supply chain determines the profit level of enterprises in BSM. Compared with VMs, the adoption of the BSM has provided BM greater profit growth. Secondly, the number of reserve batteries in the battery swapping stations and the revenue of cascade utilization are essential factors affecting the profits of battery swapping service (BSS), especially for VMs. In addition, setting a reasonable range for the pricing of BSS can achieve a win–win situation for both manufacturers.

**Keywords:** battery swapping mode; electric vehicle; cascade utilization; supply chain; power structure

## 1. Introduction

With the worldwide soaring concern for carbon emissions and environmental sustainability, the electric vehicle (EV) market share has proliferated [1–3]. According to EV-Volumes data, global EV sales reached 6.75 million units in 2021, with an increase of 108% over the same period last year, shown in Figure 1 [4]. The data from the Ministry of Industry and Information Technology of the People's Republic of China [5], MIIT for short, show that the EV market keeps elevating, with production and sales increasing by more than 160% yearly, ranking the first in the world for seven consecutive years. However, the surge in demand has led to an imbalance between the supply and demand of battery raw materials, which has led to an increasing cost of vehicle manufacture. At the same time, with technological progress and industrial improvement, the new-energy automobile subsidy standards implemented by various countries fell from 2021, both of which significantly impact end-sales and industry sustainability [6]. Faced with stubbornly high-cost pressure, the EVs industry urgently needs business model innovation and battery recycling to broaden sources of income and reduce potential loss.

Several enterprises try to adopt the battery swapping mode (BSM) to solve the problem that the cost of battery production impedes their operations from reaching the lucrative level [7]. The BSM is an innovation of the business model that batteries can be recharged while stored centrally and swapped to replenish energy quickly for the EVs in the battery swapping stations [8]. To promote the market-oriented application of EVs, the Chinese government is popularizing the BSM energetically and adding it to the government work

report as an essential part of the "new infrastructure". In the short run, the BSM can effectively relieve users' anxieties about the driving range, sharply reduce the down payment of EVs, and provide a new revenue growth chance for service providers. After the battery replacement, the battery can be charged at a constant temperature and humidity, which reduces loss and extends the battery life. In the medium and long term, the BSM can contribute to the recycling and cascade utilization of batteries and be conducive to the sustainability of the whole EV industry.

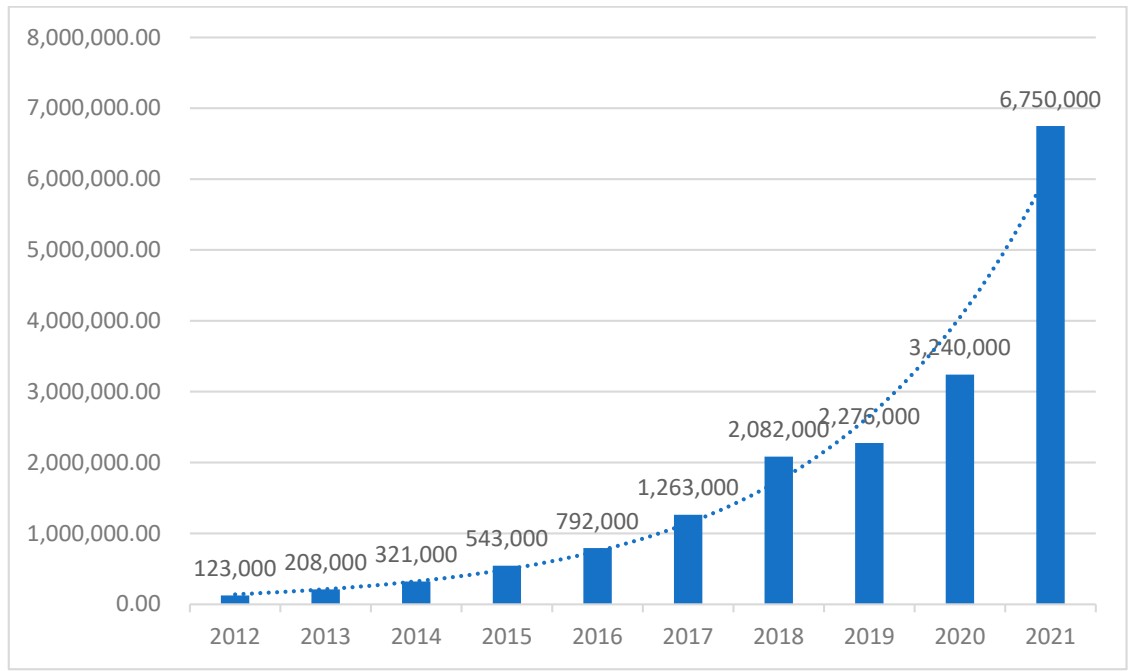

**Figure 1.** Global vehicle sales from 2012 to 2021.

For extreme integration and cost reduction, the traditional charging EV manufacturers, such as Tesla, embrace the vehicle–battery integration design solution. Meanwhile, the battery-swapping EV manufacturers, such as NIO, offer a service system to increase the driving range further, reduce the purchase threshold, improve user experience, and attract consumers with chargeable, swappable, and upgradable batteries. Recently, Contemporary Amperex Technology Co. Limited, called CATL for short, which is the imperial enterprise of battery manufacturers (BMs), has entered the battery swapping service (BSS) market with lower service costs and more abundant technology accumulation, thus creating a closed loop for critical businesses from production and manufacture, through the consumer terminal, to ultimately energy storage utilization. The deterioration of range anxiety and the rapid rise in manufacturing costs have led to the implement of the BSM and the large-scale application of BSM has caused it to be possible to unbundle the battery from the EV, which is the base of lease purchase. The unified management of batteries, during the lease purchase, creates the subsequent cascade utilization with a greater commercial value, which undoubtedly support the further promotion of the BSM. To sum up, different core enterprises, with essential roles in the industrial and innovation ecosystem, support and shape different brand tensions and competitive advantages, as shown in Figure 2. Currently, in the EV industry, there is no unified outlook on improving energy replenishment and the competitive advantages of each scenario differ considerably.

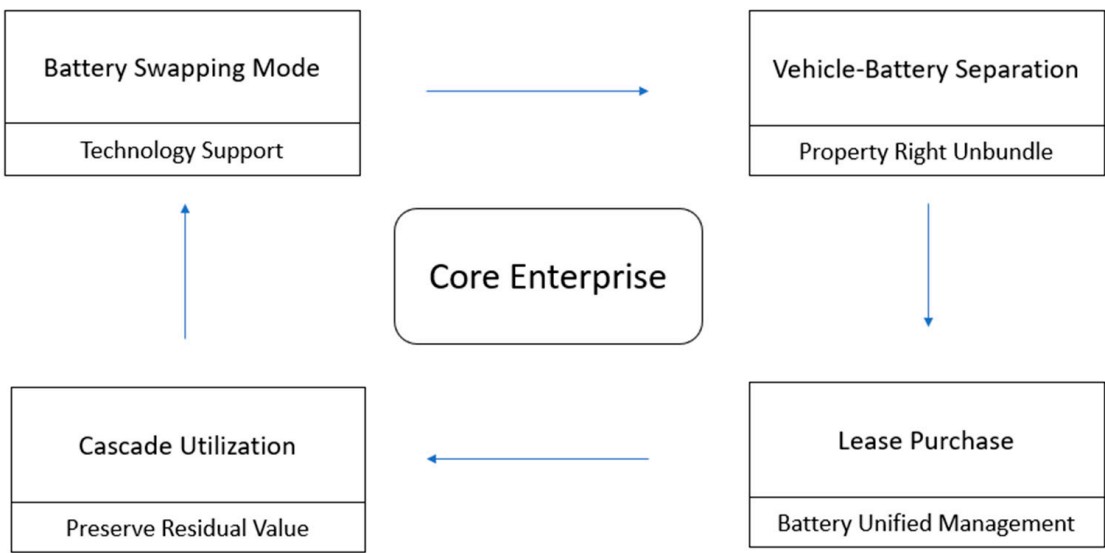

**Figure 2.** Closed-loop logic of Battery Swapping Service.

Although there are some relevant studies on the pricing decision in the supply chain of EVs under the BSM, more attention is paid to the pricing strategy of the forward supply chain. It does not take into account that the BSM affects the income attribution and major method of battery recycling in the process of causing changes in the internal power structure of the supply chain. At the same time, the supply chain led by different kinds of manufacturers contains the different characteristics of technology research and development, which will lead to different operating costs and consumer satisfactions. In real life, consumers of EVs have strong range anxieties and demand for BSS, but they also hesitate in the face of high input costs due to a lack of economies of scale and rising raw material prices. Facing the urgent needs of consumers and high operating costs, how manufacturers should formulate reasonable pricing strategies to promote the development of the market has become an important research issue. Given the situation mentioned above, based on previous studies, this paper discusses the impact of the power structure and battery cascade utilization on the manufacturer's pricing strategy, especially in the BSM, compares the differences between product pricing and overall profits in various scenarios, and provides some corresponding suggestions to solve practical problems.

The rest of the paper is organized as shown in Figure 3. Section 2 reviews the current relative literature; Section 3 describes the problems and lists the scenarios and assumptions; the Stackelberg game-theoretical decision models are formulated in differential scenarios in Section 4, including the traditional charging supply chain model for comparison (Section 4.1) and the vehicle manufacturer-led supply chain model (Section 4.2.1) compared with the battery manufacturer-led supply chain model (Section 4.2.2); the numerical and sensitivity analyses for all scenarios are discussed in Section 5; the managerial insights from this study will be summarized in Section 6; and the conclusion in Section 7.

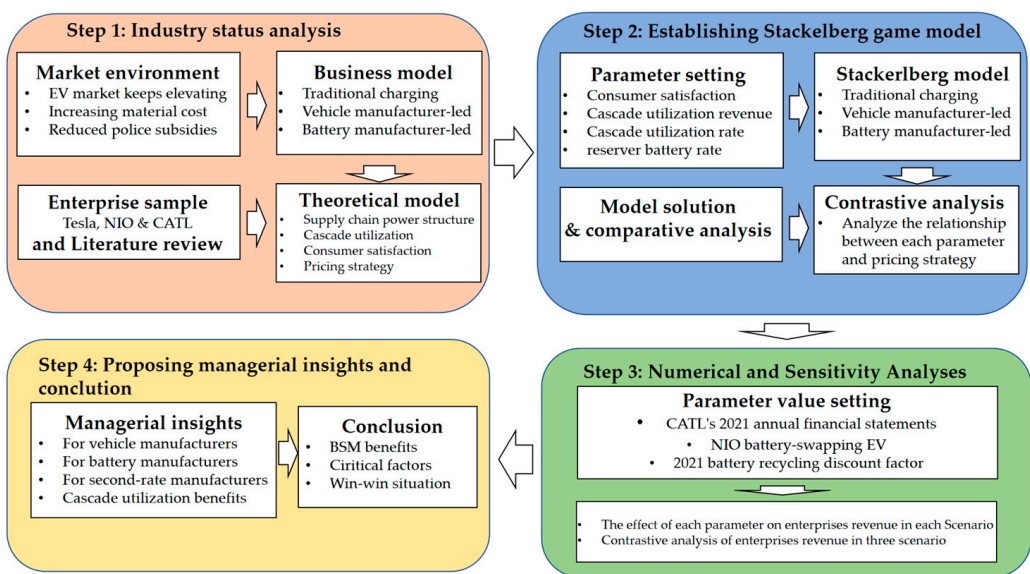

**Figure 3.** Research methodology diagram.

## 2. Literature Review

This research is closely related to the existing literature in three aspects: the operational strategy of BSM, the cascade utilization of batteries and the power structure influence on the electric vehicle supply chain. The relevant studies are reviewed below.

### 2.1. The Operational Strategy of BSM

As an innovative business model, many scholars have researched BSM but with varying degrees of focus [9–11]. Among this research, Better Place is often studied as a classic example of the failure of the BSM. Geraedo et al. argued that the key to the failure of the Better Place model was the low penetration of EVs in the global vehicle fleet in that period. Therefore, its business model innovation still had limited potential for society-wide EV penetration [12]. Benjamin K. Sovacool argued that while Better Place had many reasons for over-expansion, higher-than-expected asset costs, and overall mismanagement, it was the confluence of social, political, and other environmental factors at the heart of the problem [13]. However, it is undeniable that the business model is more sensitive to external factors, such as the operating environment and consumers [14,15], than the mere technological progress and internal optimization of management [16]. Some studies have focused on battery swapping facilities, with many of them on the site selection of facilities and operating costs. Mahoor et al. considered stochastic consumer demand for batteries and showed that demand shifting and energy sell-back could reduce the operating costs of battery swapping facilities [17]. Wu Hao reviewed the literature from the period and summarized the operating models and decision scenarios of battery swapping facilities and provided a comparative analysis of key characteristics, such as the number of battery types [18]. Fewer scholars have investigated the pricing strategy of BSM. Liang et al., 2018 concluded that peak-to-valley pricing has optimal energy efficiency and economic effects by simulating consumer responses to different battery swapping prices [19]; Liang et al., 2021 concluded that the battery cost and swapping price are the key factors affecting the net revenue of the battery system over its whole life cycle [20]. It can be seen that the optimal decision-making of relevant upstream and downstream enterprises in the supply chain of BSS are yet to be discussed and the study of which will be an effective supplement to the existing studies.

### 2.2. The Cascade Utilization of Battery

According to SNE Research, the global battery usage for EVs in 2021 reached 296.8 GWh, with a year-on-year increase of over 102%, and showed a rapid growth trend [21]. As the core

component of EVs, the battery represents over 40% of the total cost of an electric vehicle and retains a high residual value after decommissioning [22,23]. Gu et al. suggested that battery recycling and reuse would help reduce raw material consumption and environmental impact compared to new battery manufacturing [24]; Wang et al. proposed that transportation costs, carbon taxes, and the number of used batteries were the three main factors affecting the optimal design of recycling networks [25]. Zhang qi found that although establishing a system of incentives and penalties effectively increased recycling ratios, it was detrimental to the overall revenue growth, and the partnership between manufacturers and retailers was key to achieving optimal actual recycling ratios [26]. Tang et al. suggested that the advantage of the manufacturer's dominant position in the supply chain and the retailer's sales network will cause reverse logistics recycling to be more advantageous compared to other models [27].

In particular, cascade utilization has attracted more attention as a new type of recycling in this field. By utilizing residues and recycled materials, cascade utilization can efficiently use resources to expand the total availability within a given system [28,29], which fits in well with the high residual value of the batteries [30,31]. Zhang et al. emphasized that technological advances in battery technology are crucial to enhancing the profitability gained from cascade utilization and re-manufacturing and advised manufacturers should focus more on cascade utilization than re-manufacturing in the long term [32]. Abdel-Monem et al. certificated that the energy replenishment management can cover a certain degree of performance inequality between battery modules, which is highly beneficial for cascade utilization [33]. However, the current research on BSM has not been analyzed in detail for the replenishment scenario, especially the positive impact of the replenishment method on battery recycling has not been considered, and this paper adds further to this aspect.

It is noteworthy that, even on the transition to hydrogen–electric hybrid vehicles, the battery will still play a crucial role. Fragiacomo emphasized that the battery will be the key to the integration of sustainable systems with renewable energy applications via water electrolysis [34]. In the process of operation, Tolj et al. investigated that the operation in the hybrid battery and fuel cell powering mode will result in more stable driving performance [35]. Yu et al. summarizes various energy management strategies and proposed that the addition of the battery to integrated energy management can improve the performance of hybrid vehicles [36]. Therefore, even after the promotion of hydrogen–electric hybrid vehicles, the battery recycle will continue to be important.

### 2.3. The Power Structure and Game Theory in the EV Supply Chain

Considering long-standing power structures with different characteristics in the different EVs supply chains, many scholars have addressed this concept in their research and studied it using a game theoretic approach. Fan et al. demonstrated the effects of brand competition and vertical cooperation on pricing strategies and found that cooperation strategies permanently reduce vehicle prices, but suppliers cooperating with second-rate manufacturers will reduce price competition between manufacturers [37]. Zhu et al. found that when there is a Stackelberg game between upstream and downstream firms in the supply chain, the upstream manufacturers benefit from the over-orders placed, by downstream manufacturers, to avoid battery shortages [38]. In comparing the Stackelberg delay model with a simultaneous decision-making model, Ma investigated the differential impact of the Stackelberg pricing game and simultaneous pricing game and found that the Stackelberg game, for its delay factor, allows followers to adjust their pricing decisions in time, thus enhancing the stability of the structure [39]. An essential difference between the BSM and traditional charging mode is that the battery is owned by the service providers rather than the consumers, which will cause significant disruption to the value distributed in the supply chain. Therefore, the power structure and game theory of the supply chain, which have a significant impact, should be considered in the study.

### 2.4. The Impact of the Supply Chain System on Environmental Issues and Customer Satisfaction

As a transition vigorously promoted by the whole society, many scholars have studied the main obstacles and driving forces of electric mobility [40,41]. Among them, environmental issues score highly, based on which EV manufactures implement corresponding measures. Gu et al. evaluated a closed-loop supply chain with battery recycling and proved that the EV market will be negatively impacted by the lack of manufacturer incentive and awareness to recycle used batteries [42]. Ashok et al. proposed that awareness improvement programs on environmental benefits can motivate the consumers to promote the EV transition even though several factors including purchasing cost, driving range, range anxiety, and lack of charging infrastructure still impede the transition [43]. Through the comprehensive review of barriers to the adoption of EVs, Chidambaram et al. generalized that the barriers are related to the battery technology and that battery optimization will be the key to raise EV technology acceptance in which consumer mind-sets plays a pivotal role [44]. Hu et al. used a discrete-event simulation approach to simulate a car-sharing program that discovered that limited battery capacity will seriously reduce consumer satisfaction as well as the vehicle utilization rate [45]. Hence, as an important factor affecting consumer satisfaction and purchase demand, the technology level of battery and energy replenishment will be emphatically considered in the demand function.

### 2.5. Summary

The above research shows a lack of research on the analysis of BSM combined with battery recycling based on the different power structures within different supply chains. In reality, the network of BSS has been gradually launched in first-tier cities with inadequate charging facilities and a lack of private charging facilities, and the resulting cascade utilization process has gradually matured. The power structure significantly influences how enterprises carry out BSS and the value distribution in the supply chain. There is a lack of adequate theoretical research and academic recommendations in this area, while the service has emerged on a large scale. Therefore, this article establishes three models for the supply chain of EVs, referring to the supply chains of Tesla, NIO, and CATL, considering the cascade utilization of batteries, in order to study the impact of BSS, cascade utilization technology, supply chain power structure, and other parameters on the enterprises involved in the supply chain, discuss the influence of BSM, and make recommendations for other enterprises to promote the BSM further.

## 3. Model Formulation and Analysis

### 3.1. Background Description

Currently, there is a widespread power imbalance between upstream and downstream manufacturers in the EV supply chain triggered by the different competitive advantages that vehicle manufacturers (VMs) and BMs have in the process of industrial development. Containing different power structures in different supply chains, this paper will use the Stackelberg game to establish three supply chain game models with different leaders, as shown in Figure 4.

In the traditional charging supply chain (Scenario T), the VM is the leader in the supply chain, while the BM is the follower. The VM buys the batteries from the BM, sells the vehicle with the battery together to the consumer after assembly, and recycles the battery when it is decommissioning, paying the consumer a lower price. Due to the lack of persistent professional testing of batteries during charging and replenishment, unknown capacity loss after long-term use, and lack of daily operation data, the recycling method is mainly based on dismantling and recycling precious metal materials. Cascade utilization can be barely applied in this scenario for significant detection difficulty and time costs.

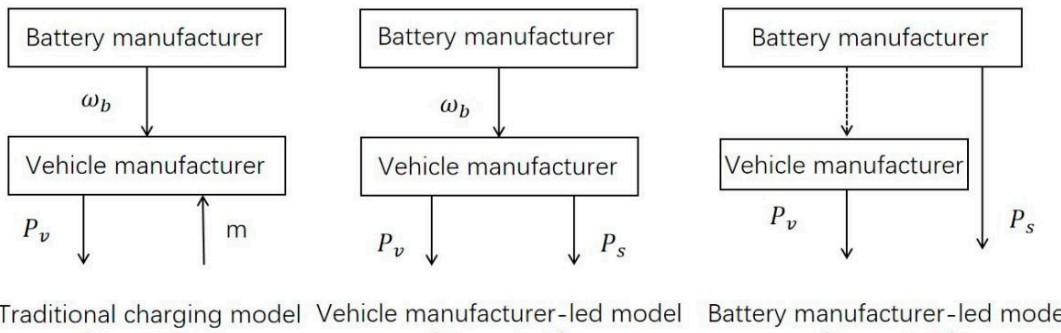

**Figure 4.** Supply chains in differential power structures.

In the vehicle manufacturer-led supply chain (Scenario V), the VM is responsible for designing and developing the battery swapping technology and the BM produces customized batteries according to the design requirements. The VM no longer sells the vehicles with the battery but only sells the body parts of the vehicle and provides the BSS as a substitute, which is the long-term lease.

In the battery manufacturer-led supply chain (Scenario B), the VM is only responsible for manufacturing the body part of the vehicle. At the same time, the BM provides the BSS to consumers jumping across the original supply chain hierarchy and takes responsibility for battery recycling. This scenario will be utterly devoid of revenue from the sale of batteries as the batteries assembled in the EVs will be returned to the battery swapping station through the progress of battery replenishment.

*3.2. Basic Assumption*

Before establishing the mathematical model, the following assumptions are adopted regarding the reality. First and foremost, the battery lease is considered a prerequisite for the BSS in this paper. As the necessary tests will be conducted during each replenishment, the battery will be promptly transferred to the repair or decommissioning process if any problems are found. Therefore, the cascade utilization ratio should be a larger percentage than the completely scrapped ratio. As the decommissioned batteries will still be dismantled and recycled after the cascade utilization, the overall benefits generated should be greater than those obtained by the dismantling and recycling material. The number of reserve batteries required for the BSS can be estimated and, according to the operation of the NIO battery swapping station, a ratio of less than 1:1.3 of EV sales can be used to meet the operational demand [46]. To simplify the calculation, consumers will only use a single replenishment method. Additionally, the investment in EV battery and replenishment technology will be regarded as a one-off expenditure, the impact of which on the production cost will be negligible.

*3.3. Demand Function Construction*

In this paper, the EVs are treated simply as a combination of a vehicle body and a battery, thus considering only the secondary supply chain consisting of a BM and a VM. The body of the vehicle is designed and manufactured by the VM, while the battery is designed and manufactured by the BM.

In this paper, we regard the demand for EVs as the sum of a linear function of the price consumers offered, including the unit price of the EV including a vehicle body and a battery, the price of energy replenishment, and the technology preference. The consumers have a coefficient for the unit price of an electric vehicle, which is a one-off transaction and represented by $a$, $0 < a < 1$. $b$ represents the cross-price elastic coefficient of the BSS price paid by installments, $-1 < b < 0$, so the absolute value of $a$ should be greater than the one of $b$, which means consumers are more willing to accept installments facing the same price. $P_C$ and $P_S$ denote the unit price of an electric vehicle without the battery and the unit

revenue of the life-cycle battery swapping service, respectively. Additionally, considering that battery and energy replenishment technology is the key to user experiences and there is a certain technology preference represented by $\theta$ among consumers [47,48], $h$ represents the technology level of battery and energy replenishment. Referring to the literature [49], the technology research and development cost is set as $I = \frac{1}{2}g_ih^2$. The demand function in this paper is as follows.

$$Q = \phi - a(P_v + \omega_b) + bP_s + \theta h \tag{1}$$

In reality, battery-swapping vehicles can still use several charging methods, including fast-charge, so the technology research and development of Scenario T will be included in that of the BSM. Therefore, the technology research and development cost coefficient of Scenario T should be smaller than that of any BSM because Scenario T does not require research into so many technologies, thus $g_1 < max\{g_2, g_3, g_4\}$. At the same time, considering that BMs have accumulated certain technology in battery pack development and charging and other related fields, to achieve the same level of technology, the technology research and development cost coefficient of the VM should be greater than that of the BM ($g_2 > g_3$). Regardless of any scenario, the cost of EV battery and replenishment technologies is an enormous numerical value.

*3.4. Nomenclature*

To show the meaning of the symbols used in this paper clearly, the notations are defined and summarized in Table 1.

**Table 1.** The description of the symbols.

| Symbol | Symbol Definition |
| --- | --- |
| $Q$ | demand for EV in the market |
| $\phi$ | potential market size |
| $a$ | elastic coefficient to the price of one-off transaction |
| $b$ | cross-price elastic coefficient to the price of battery lease ($b < 0$) |
| $C_v$ | unit cost of producing a body of the vehicle |
| $C_b$ | unit cost of producing a new battery with the raw materials |
| $C_o$ | unit cost of operating the battery in the swapping station |
| $g_i$ | technology research and development cos coefficient in each scenario ($g_2 > g_3 > g_1$) |
| $\theta$ | consumers' sensitivity for the technology of battery and replenishment |
| $V_a$ | the revenue of battery disassemble recycling |
| $V_b$ | the revenue of battery cascade utilization ($V_a < V_b$) |
| $f$ | the ratio of the decommission batteries recycled with cascade utilization ($f < 1$) |
| $\lambda$ | the ratio of the actual battery quantity needed to the demand for EV in the market ($\lambda > 1$) |
| $m$ | unit recycle price for decommissioned battery from the consumer |
| $k$ | the ratio of the life-cycle BSS revenue to the battery wholesale price ($k > 1$) |
| $\pi_V$ | the profit of vehicle manufacturer |
| $\pi_B$ | the profit of battery manufacturer |
| $P_v$ | unit price of an electric vehicle body without the battery |
| $P_s$ | unit revenue of battery swapping service during the whole life cycle |
| $\omega_b$ | unit wholesale price of the battery from the battery manufacturer |
| $h$ | the technology level of battery and energy replenishment |

## 4. Stackelberg Game Model Analysis

*4.1. Scenario T*

Firstly, the traditional charging model is constructed as the comparison scenario. In this model, consumers only use direct charging to replenish energy, therefore $Ps = 0$. The VM bears the primary responsibility for battery recycling, the profit functions of the VM and the BM are shown as follows and the decision sequence is shown in Figure 5.

$$Q = \phi - a(P_v + \omega_b) + \theta h \tag{2}$$

$$\pi_V = Q(P_v - C_v) + (V_a - m)Q \tag{3}$$

$$\pi_B = Q(\omega_b - C_b) - \frac{1}{2}g_1h^2 \tag{4}$$

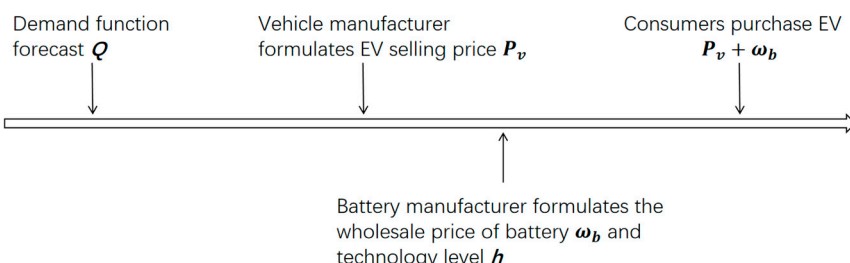

**Figure 5.** Decision sequence diagram in Scenario T.

Because the VM is the Stackelberg leader, it should be begun by characterizing the best-response function of the BM. According to the backward induction method, the optimal values of the scenario T can be derived. Because of the space limit, the proofs are listed in Appendix A and the analytical results are summarized in Table 2.

### 4.2. Analysis of BSM Supply Chain

Compared to the traditional charging model, the emergence of the BSS has intensified the competition for supply chain dominance between VMs and BMs. Only the core enterprises can integrate upstream and downstream resources in the supply chain to carry out a diversified BSS, including battery leasing and on-demand battery update, based on their original business, thus having the pricing power and disposal power of decommissioned batteries. As consumers do not need to purchase batteries in the BSM, the wholesale price of batteries no longer directly affects consumer demand, so the demand function is updated as follows.

$$Q = \phi - aP_v + bP_s + \theta h \tag{5}$$

#### 4.2.1. Scenario V

The VMs purchase batteries with a total volume of $\lambda Q$ from the BM, where $Q$ is the number of batteries required for vehicle assembly and $(\lambda - 1)Q$ is the number of reserve batteries required for the BSS. In this scenario, the wholesale price of batteries is an essential reference for the pricing of the BSS, which satisfies the relationship equation $P_s > \omega_b > C_b$. Hypothesizing $P_s = k\omega_b$, this paper discusses various pricing strategies by adjusting parameter k and the decision sequence is shown in Figure 6.

$$\pi_V = Q(P_v - C_v) + Q(P_s - C_o) - \omega_b\lambda Q + V_a(1 - f)\lambda Q + V_b f\lambda Q - \frac{1}{2}g_2h^2 \tag{6}$$

$$\pi_B = \lambda Q(\omega_b - C_b) \tag{7}$$

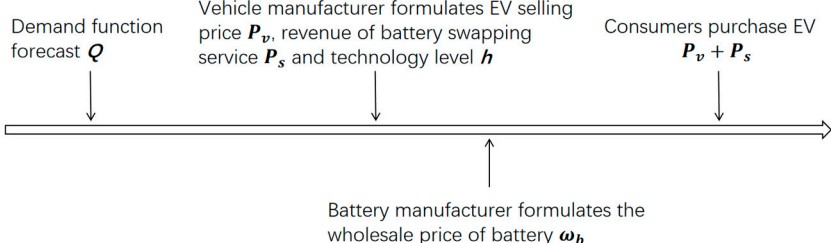

**Figure 6.** Decision sequence diagram in Scenario V.

**Table 2.** Analytical results of three Stackelberg game-theoretical decision models.

| | Scenario T | Scenario V * | Scenario B |
|---|---|---|---|
| $P_v$ * | $\dfrac{\phi + a(m + C_v - C_b - V_a)}{2a}$ | $\dfrac{bk\big(2bC_bg_2k - \theta^2((\lambda - k)C_b + M) + 2g_2\phi\big) + 2ag_2(\phi(k - \lambda) + bkM))}{4abg_2k - bk\theta^2 + 2a^2g_2(k - \lambda)}$ | $\dfrac{4bg_3\phi + a\big(3\theta^2 + 10bg_3\big)C_v - 4b^2g_3(V_a(1 - f)\lambda + V_bf\lambda - C_o - \lambda C_b)}{4a(\theta^2 + 4bg_3)}$ |
| $\omega_b$ * | $\dfrac{g_1\phi + ag_1(V_a + C_v + C_b - m) - \theta^2 C_c}{4ag_1 - 2\theta^2}$ | $\dfrac{b^2k^2(-bkC_b\theta^2 + ag_2(3bkC_b - \phi) + a^2g_2((k - \lambda)C_b + M)}{4abg_2k - bk\theta^2 + 2a^2g_2(k - \lambda)}$ | / |
| $P_s$ * | / | $\dfrac{b^2k^3(-bkC_b\theta^2 + ag_2(3bkC_b - \phi) + a^2g_2((k - \lambda)C_b + M)}{4abg_2k - bk\theta^2 + 2a^2g_2(k - \lambda)}$ | $\dfrac{g_3(2aC_v - 2\phi) - (\theta^2 + 2bg_3)(V_a(1 - f)\lambda + V_bf\lambda - C_o - \lambda C_b)}{\theta^2 + 4bg_3}$ |
| $h$ * | $\dfrac{\theta[\phi + a(V_a - m - 3C_v + C_b)]}{4ag_1 - 2\theta^2}$ | $\dfrac{bk\theta(bkC_b + \phi - a((\lambda - k)C_b + M)}{4abg_2k - bk\theta^2 + 2a^2g_2(k - \lambda)}$ | $\theta\,\dfrac{aC_v + b(V_a(1 - f)\lambda + V_bf\lambda - C_o - \lambda C_b) - \phi}{\theta^2 + 4bg_3}$ |

* For ease of presentation, we set $C_v + C_o - \lambda(fV_b + V_a - fV_a) = M$ in Scenario V.

After calculation, the optimal values of Scenario V can be derived as follows. The detailed proofs can be seen in Appendix B. For the ease of presentation, we set $C_v + C_o - \lambda(fV_b + V_a - fV_a) = M$ and the analytical results are summarized in Table 2.

**Proposition 1.** *Under vehicle manufacturer dominance, the vehicle's selling price decreases with the increase in the price of the BSS to the wholesale price of batteries when $k < 2\lambda$. See Appendix C for detail proofs.*

Proposition 1 shows that when the wholesale price of the battery is fixed and $1 < k < 2\lambda$, the selling price of the EV will fall to a certain extent as the price of the BSS increases, while, when $k > 2\lambda$, the selling price of the whole vehicle will then increase. It can be seen that the BSS is a complementary production to the battery-swapping EVs in the BSM. If the service price rises, the vehicle manufacturer will reduce the price to increase consumers' willingness to buy. If the service price breaks through the reasonable range, market demand will fall sharply, and VMs will have to sell vehicles at a higher price to compensate for the loss, thus entering a negative feedback loop.

**Proposition 2.** *The vehicle selling price will decrease with increasing cascade utilization gains when $a + 2b < 0$ in Scenario V. See Appendix D for detail proofs.*

Proposition 2 shows that when a and b meet the particular condition mentioned above, the vehicle's selling price will drop to a certain extent as the proportion of decommissioning batteries that can be recycled and the benefits of the utilization increase. The VM compensates for the high initial battery input cost required for the BSS to a certain extent through the cascade utilization of batteries. Consequently, the VM can attract consumers with a lower purchase threshold by lowering the vehicle price. Therefore, establishing a reverse recycling supply chain for decommissioning batteries catalyses the overall supply chain and promotes society-wide EV penetration.

### 4.2.2. Scenario B

In this scenario, the BM acts as the supply chain leader for conducting BSS business. The cost of the reserve battery used for the service is the production cost of batteries, the decision sequence is shown in Figure 7, and the rest is similar to scenario V.

$$\pi_V = Q(P_v - C_v) \tag{8}$$

$$\pi_B = (P_s - C_o)Q - \lambda C_b Q + V_a(1 - f)\lambda Q + V_b f \lambda Q - \frac{1}{2}g_3 h^2 \tag{9}$$

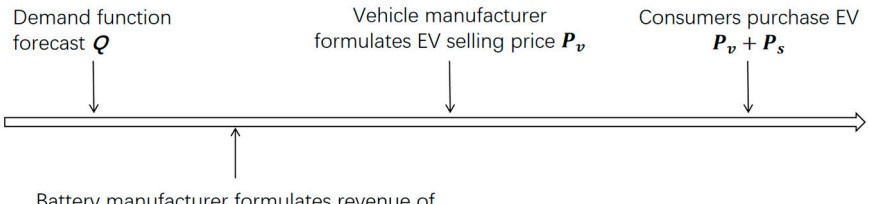

**Figure 7.** Decision sequence diagram in Scenario B.

After calculation, the optimal values of Scenario V can be derived as follows. The detailed proofs can be seen in Appendix E and the analytical results are summarized in Table 2.

**Proposition 3.** *The price of BSS decreases as the proportion of batteries that can be recycled within cascade utilization increases, while battery technology development increases accordingly.*

The proof process is similar to proposition 2 and is therefore omitted. Proposition 3 shows that consumers can access lower-priced BSS as the proportion of batteries that can be recycled within cascade utilization increases. At the root of this, BMs can fully utilize the residual value of batteries through cascade utilization, thereby not fearing the high upfront investment costs for the additional batteries required for the BSS. By lowering the price of the service and improving the level of technology research and development, consumers are attracted to choose the service while ensuring that they do not lose money.

**Proposition 4.** *The profits of both VMs and BMs increase with the proportion of batteries that can be recycled within cascade utilization.*

The proof process is similar to proposition 2 and is therefore omitted. Proposition 4 shows that as the proportion of decommissioning batteries that can be recycled within cascade utilization increases, both upstream and downstream supply chain companies can obtain higher profits. Combined with proposition 3, the cascade utilization of the decommissioning batteries can effectively reduce the price of the BSS while improving technological research and development, thus attracting consumers to purchase battery-swapping EVs. The BMs, initially battery suppliers, solve consumers' problems by providing cross-level services directly for consumers, strengthening technology development, and generating higher profits while driving the sales of battery-swapping EVs, thus realizing value co-creation with VMs.

## 5. Numerical and Sensitivity Analyses

According to CATL's 2021 annual financial statements [50], the average cost of a standard 100 KWh size battery is about 50,000 CNY and, after referring to the 2021 battery dismantling and recycling discount factor [51], the dismantling and recycling revenue is set at 40,000 CNY. Using the case of a hot-selling NIO battery-swapping EV for an example, the selling price of the vehicle body without the battery is around 280,000 CNY and the manufacturing cost accounts for about 40% of the vehicle price after excluding taxation and operation and sales costs, so the manufacturing cost of which in this paper is 110,000 CNY. According to Lih et al.'s study [52], cascade utilization will provide about 35% profit, so the total benefit is set at 60,000 CNY. Considering the actual operation of NIO battery swapping stations and the theoretical projections, this paper assumes $\lambda$ as 1.2 to simulate a more mature and larger BSS market than it is today. All parameters in this paper are adopted as shown in the following Table 3.

**Table 3.** Parameter value setting.

| Symbol | Symbol Definition | Value Setting |
|:---:|:---:|:---:|
| $\phi$ | potential market size | 2,500,000 |
| $a$ | elastic coefficient to the price of one-off transaction | 0.5 |
| $b$ | cross-price elastic coefficient to the price of battery lease ($b < 0$) | $-0.45$ |
| $C_v$ | unit cost of producing a body of the vehicle | 110,000 |
| $C_b$ | unit cost of producing a new battery with the raw materials | 50,000 |
| $C_o$ | unit cost of operating the battery in the swapping station | 100 |
| $g_1$ | technology research and development cost coefficient in Scenario T | 100 |
| $g_2$ | technology research and development cost coefficient in Scenario V | 160 |
| $g_3$ | technology research and development cost coefficient in Scenario B | 140 |
| $\theta$ | consumers' sensitivity for the technology of battery and replenishment | 0.3 |
| $V_a$ | the revenue of battery disassemble recycling | 40,000 |
| $V_b$ | the revenue of battery cascade utilization | 60,000 |
| $f$ | the ratio of the decommissioned batteries recycled with cascade utilization | 0.95 |
| $\lambda$ | the ratio of the actual battery quantity needed to the demand for EVs in the market | 1.2 |
| $m$ | unit recycle price for decommission battery from the consumer | 25,000 |

### 5.1. The Effect of the Pricing Strategy on Enterprise Revenue in Scenario V

In Scenario V, the ratio of the life-cycle BSS revenue to the battery wholesale price is an essential reference value for the pricing of the BSS. From the consumer's perspective, the two are the one-off purchase costs of a rechargeable EV and the long-term replenishment cost of purchasing a battery-swapping EV, respectively. From the perspective of the VM, the two are the revenue and cost of the battery used for the BSS, respectively. The pricing strategy of the VM can be reflected by adjusting the ratio of the above two.

From Figure 8, it can be seen that the ratio of the price of the BSS to the wholesale price of the battery and the profits of both upstream and downstream enterprises in the supply chain are closely related in Scenario V. When $1 < k < 2.4$, the battery manufacturer's profit proliferates with the growth of K and the vehicle manufacturer's profit decreases slowly at this time. The VM can attract consumers to buy the vehicle through the low price of replenishment service and thus obtain higher revenue from the sale of the vehicle, but the lower price of the service reduces the profit margin of the BM. When $k > 2.4$, the overall market demand is sluggish due to the high replenishment cost, so the supply chain's overall profit decreases rapidly as K increases. It can be seen that the increase in the price of the BSS, as a complementary production to the battery-swapping vehicle, will lead to an increase in the cost of using the battery-swapping vehicle, which in turn will reduce consumers' willingness to purchase the vehicle. The VMs should consider the impact of the pricing of BSS on their own vehicle sales and the profits of upstream suppliers when they dominate the Stackelberg game.

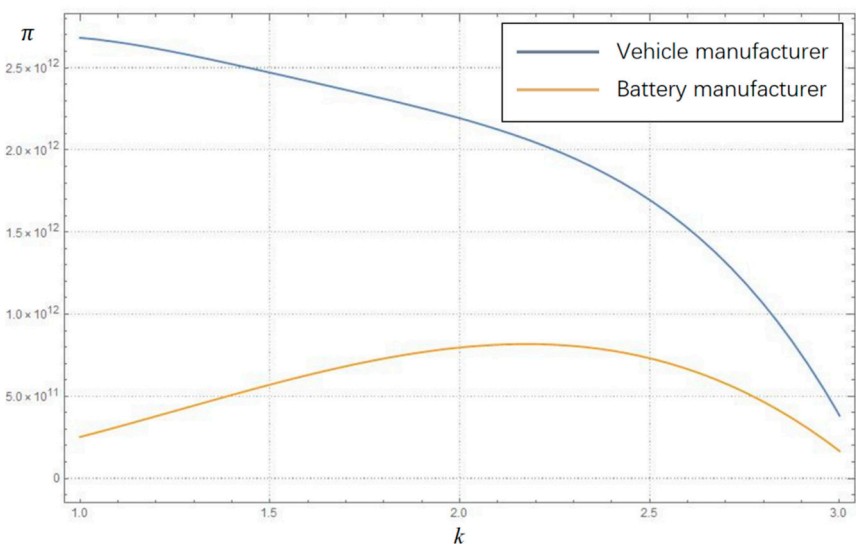

**Figure 8.** The effect of the pricing strategy on enterprises revenue in Scenario V.

### 5.2. The Effect of the Reserve Battery Quantity on Enterprises' Revenue

The purchase price of reserve batteries in a battery swapping station differs for the VMs and BMs. In the absence of revolutionary improvements in battery replenishment technology, the number of reserve batteries in the battery swapping stations limits the user experience to a certain extent. This sub-section examines the impact of the number of reserve batteries on the business's profitability.

Figure 9 shows that the VM profit in Scenario V decreases with the increasing number of reserve batteries in the battery swapping station. In contrast, the BM in Scenario B is not sensitive to the increasing ratio, for which the reserve batteries are a significant expenditure for the BSS, and the number of reserve batteries dramatically affects the company's profitability. The cost of batteries used by VMs is much higher than that of BMs, so the latter is more sensitive to the increasing number of reserve batteries.

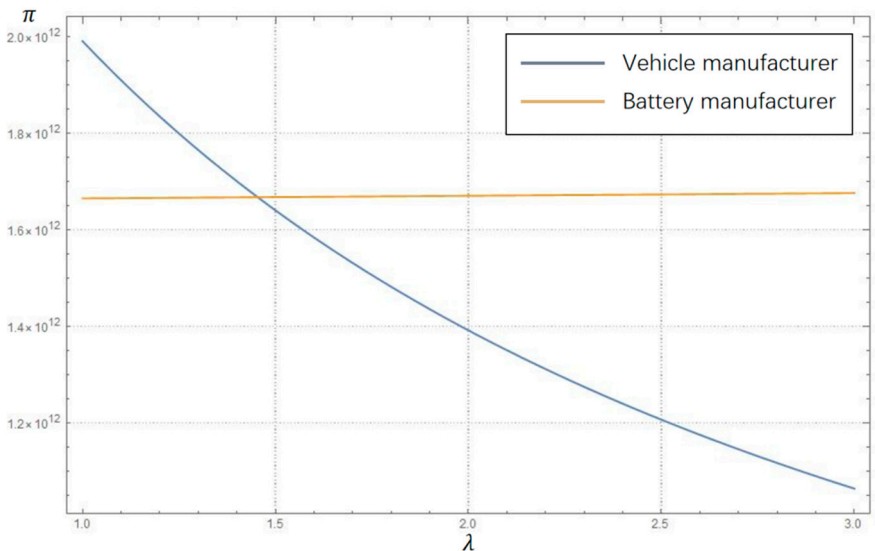

**Figure 9.** The effect of the reserve battery quantity on BSS providers revenue.

From this perspective, BMs have a competitive advantage when they promote the BSS. BMs can increase service levels by setting up more facilities in the same service area or storing more batteries in a single facility with lower costs. For VMs, it is essential to use new digital tools such as mega data analytics to predict the spatial and temporal distribution of consumer demand for battery replenishment to improve service levels with minimizing costs. If the charging requirements can be guided and transferred between battery swapping stations, orderly charging and load shaping will be helpful to reduce the reserve battery quantity.

*5.3. The Effect of Cascade Utilization Ratio on Recycling Revenue*

The role of cascade utilization in the BSM should be provided enough importance, including the cascade utilization ratio and income, both of which will cover a large proportion of the expenditure.

As seen from Figures 10 and 11, the overall revenue of the supply chain in both cases increases with the proportion of batteries that can be recycled within cascade utilization and the revenue from that. Compared to Scenario T, where it is challenging to collect decommissioned batteries, the recycling in the BSM has advantages in all aspects. However, in reality, the main field of cascade utilization is energy storage, such as clean energy storage, peak load shifting, etc. Currently, China's energy storage market has not yet been fully developed and, overall, is still in the demonstration application stage. The rational point of view in the short-term cascade utilization income is challenging to enhance significantly.

Combined with the analysis of reserve batteries, in addition to market demand, reserve batteries are a significant expenditure for enterprises, and how to increase the proportion of batteries that can be recycled as far as possible in a limited number of batteries should be the core issue for service providers. They should pay attention to the detailed testing of batteries during the replenishment to lay a solid foundation for the cascade utilization business and thus form an effective battery life-cycle management. Currently, conventional testing in the station is achieved by connecting the battery management system. If more testing means can be added to the facilities or if the sample of batteries used under high loads can be inspected in detail, that will play an essential positive role in optimizing the whole life-cycle management. By establishing a database and simulation model of the attribute parameters of decommissioning batteries, using mega data and other technical means to design a safe, reliable, efficient, and highly compatible system for the cascade scenario, the effective management of decommissioning batteries can be achieved.

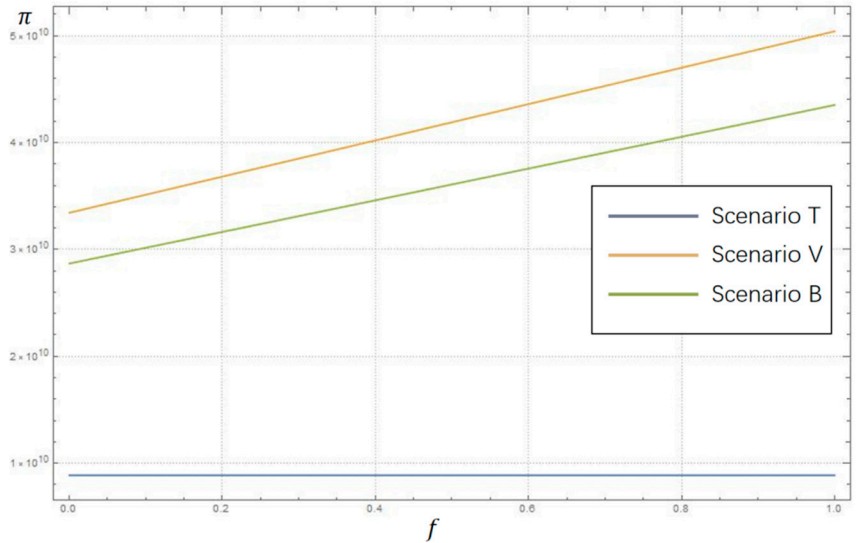

**Figure 10.** The effect of cascade utilization ratio on recycling revenue.

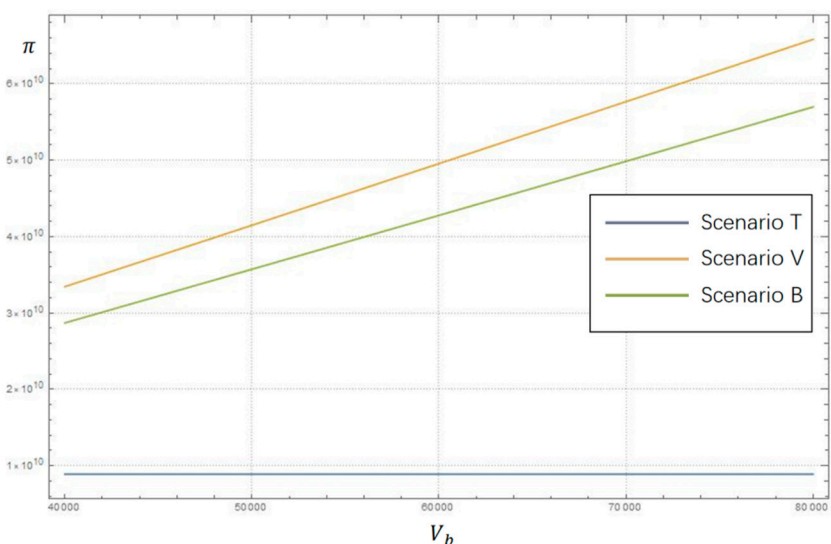

**Figure 11.** The effect of cascade utilization income on recycling revenue.

*5.4. The Effect of Elastic Coefficient on Enterprises Revenue*

As there are currently two products that affect the profitability of the business, the different market conditions and price sensitivities of users will affect the profitability of the business.

As seen from Figure 12, in all cases, the profit of the VM decreases gradually as the elastic coefficient $a$ increases. As seen from Figure 13, in the case of vehicle manufacturing developed as the main business, the profit does not change significantly with the change of $b$. However, in Scenario V, a significant increase occurs as the absolute value of $b$ decreases. Overall, Scenario V is much more profitable than the other models.

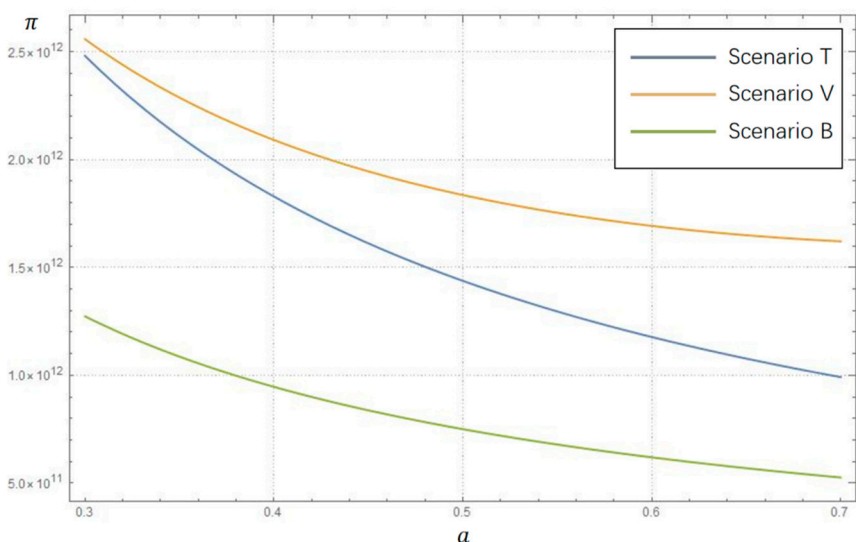

**Figure 12.** The impact of elastic coefficient a on VM revenue when we set $b = -0.45$.

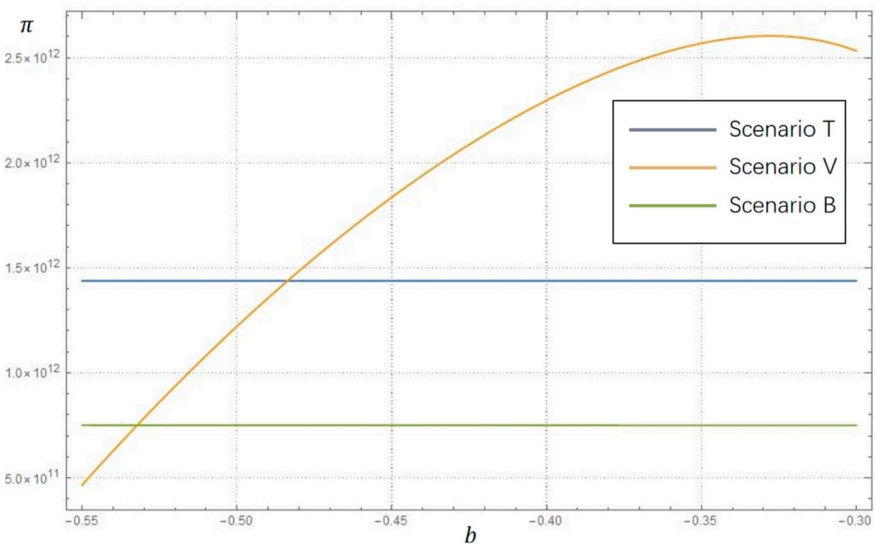

**Figure 13.** The impact of cross-price elastic coefficient b on VM revenue when we set $a = 0.5$.

The comparison shows that the BSS significantly enhances the profitability of the VM. However, the pricing decision still needs to be considered in the process of promoting the BSS. In Scenario V, the high cost of providing the service is due to the fact that the VM does not have the capacity to produce batteries. In the early stages of promotion, consumers are relatively sensitive to the replenishment pricing. At the beginning, the service providers are using a penetration pricing strategy to attract consumers and then in the frame of the market scale they gradually raise prices to achieve benign development. For the VM, there is no doubt that their profits will be reduced in the initial stages of adopting this pricing strategy. With the deepening of the promotion, consumers will gradually become less sensitive to the price of the BSS. They will instead pursue other factors such as service experience, which will result in higher profits for the vehicle manufacturers who have the advantage of the terminal brand by virtue of the scale effect.

As can be seen from Figure 14, the BM's profit fluctuates the most in Scenario T. The sensitivity of consumers to the wholesale price of EVs significantly affects the BM and the profit gradually decreases as the elastic coefficient $a$ increases. In Scenario V, the marginal profit of the VM from the sale of the vehicle body drops significantly as the elastic coefficient $a$ increases and the loss of profit from the reduction in the price of the vehicle body can

be compensated by an appropriate increase in the pricing of the BSS, thus improving the profitability of the BM. As seen from Figure 15, for the BM, although Scenario T and Scenario V have their advantages and disadvantages, they are both significantly lower than Scenario B.

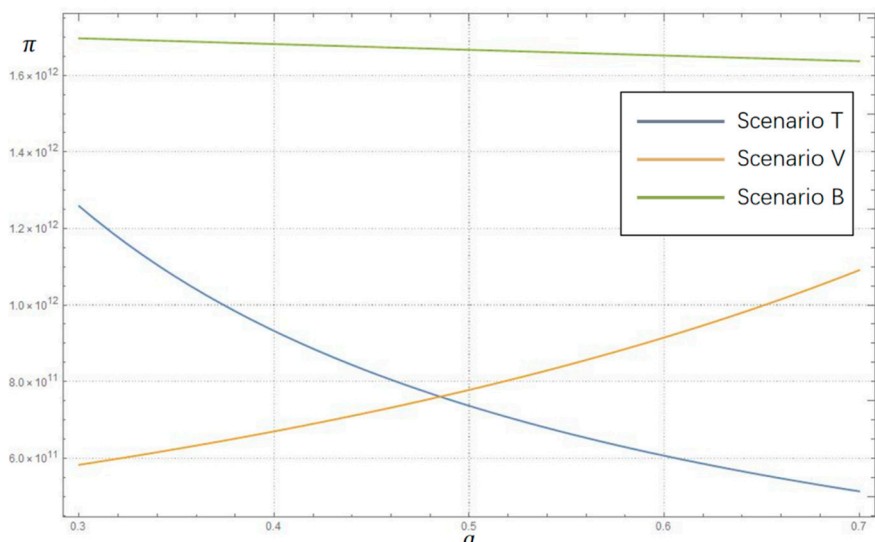

**Figure 14.** The impact of elastic coefficient *a* on BM revenue when we set *b* = −0.45.

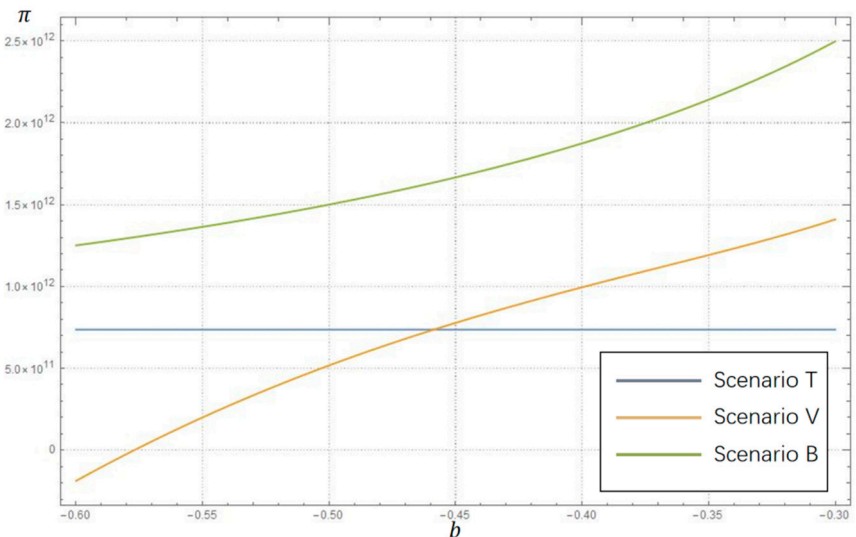

**Figure 15.** The impact of cross-price elastic coefficient *b* on BM revenue when we set *a* = 0.5.

Although BMs hold the core component of EVs, they are far away from the consumer terminals of the supply chain. They lack brand advantages due to the production characteristics, which limits their bargaining space to the bargaining power of the VMs. Currently, most BMs, through years of technology accumulation, firmly master the key aspects in the supply chain of battery manufacturing and recycling. Their business sustainability will be significantly enhanced if they complete their core business of closing the loop by carrying out BSS. Participation in the BSS market will boost overall supply chain profits, while the profit of BMs will increase significantly. The BMs aiming to transform into this market should compete for supply chain dominance while maintaining strategic cooperation with vehicle manufacturers, collaborating on research and development, and sharing costs to amplify their cost advantages further.

### 5.5. Contrastive Analyses of Variables in Differential Power Structures

The contrast of both sides of the business in the three power structures are presented in Figures 16–18.

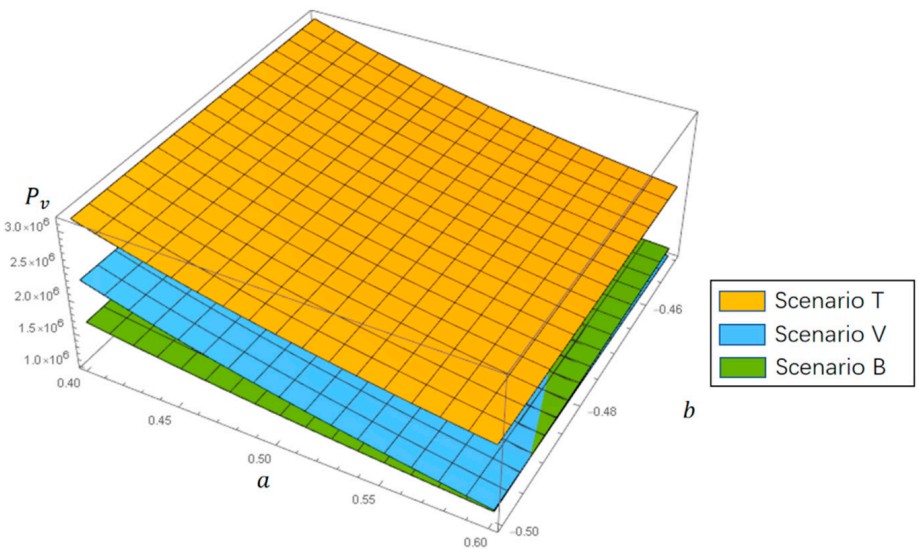

**Figure 16.** The contrast of electric vehicle body unit price $P_v$ in three scenarios.

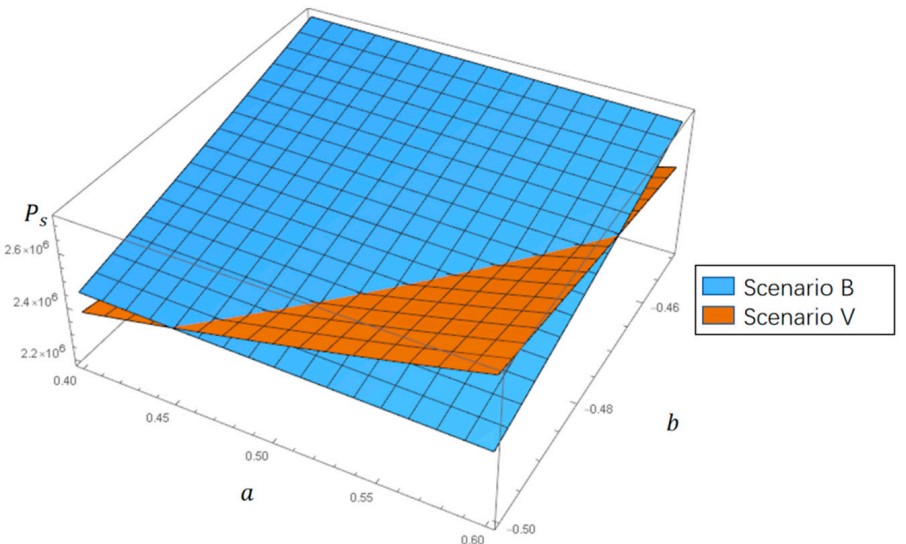

**Figure 17.** The contrast of battery swapping service revenue $P_s$ in three scenarios.

The comparison shows that the BSM plays a significant role in reducing the unit price of electric vehicle bodies without batteries, which means that consumers' purchase thresholds can be reduced highly. Compared with Scenario T, the unit price of EV bodies under the two scenarios of the BSM decreases by 15%–50%. The comparison between the two BSM scenarios shows that the unit price of the EV bodies and the price of the BSS have their own advantages and disadvantages that depend on the market circumstances.

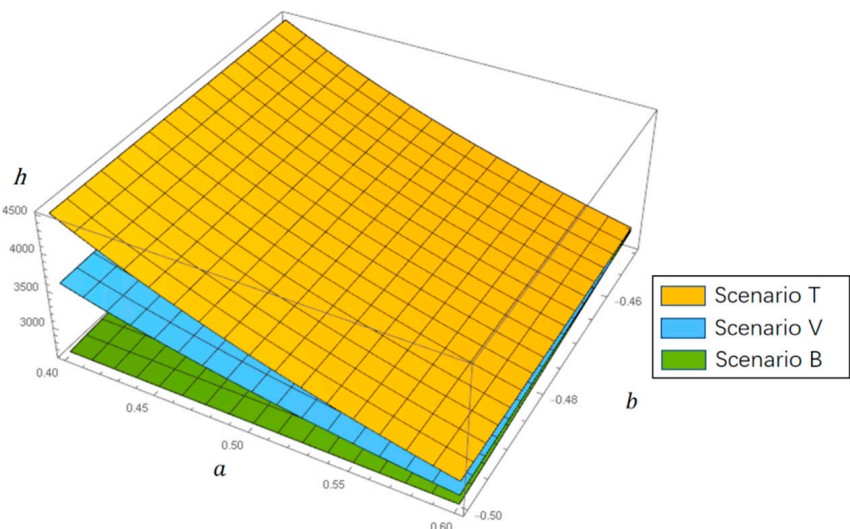

**Figure 18.** The contrast of technology level *h* in three scenarios.

On the other hand, the technology level of battery and energy replenishment differs from each scenario. The battery swapping service, compared with the traditional charging, requires more technical investment, which causes the vehicle manufacturers, who keep the revenue of EV body sales in all scenario, to be more capable of increasing the technology level than battery manufacturers. The technology research and development cost coefficient of Scenario T should be smaller than that of any BSM because Scenario T does not require research into battery swapping technologies, which makes it far superior to all other scenarios. The comparison of the optimal decision shows that, if the same technology investment will be maintained as other competitors, it will pose a potential threat to the corporate profits of battery manufacturers.

*5.6. Analyses of Enterprises' Revenue in Differential Power Structures*

The profits of both sides of the business in the three power structures are presented in Figures 19–21.

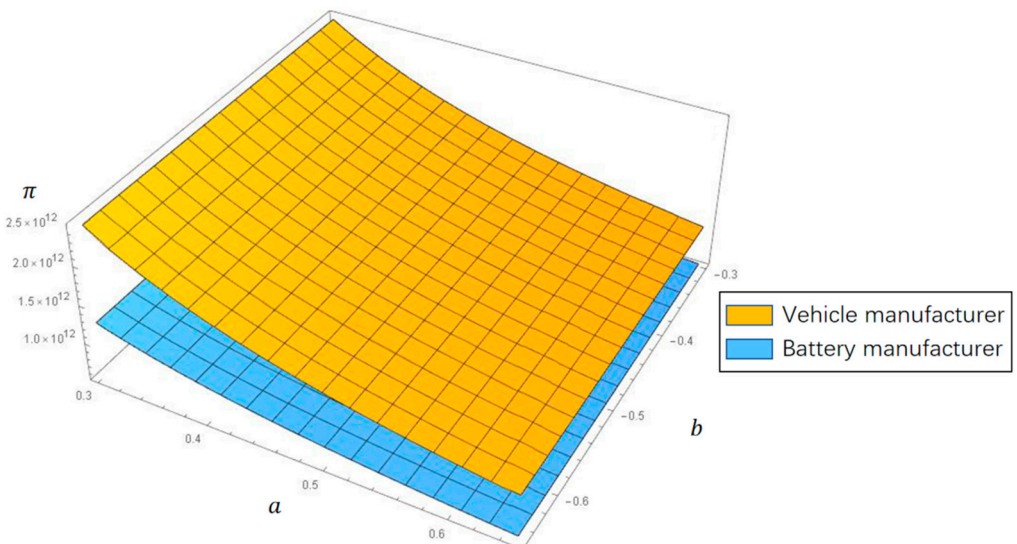

**Figure 19.** The revenue of both enterprises in Scenario T.

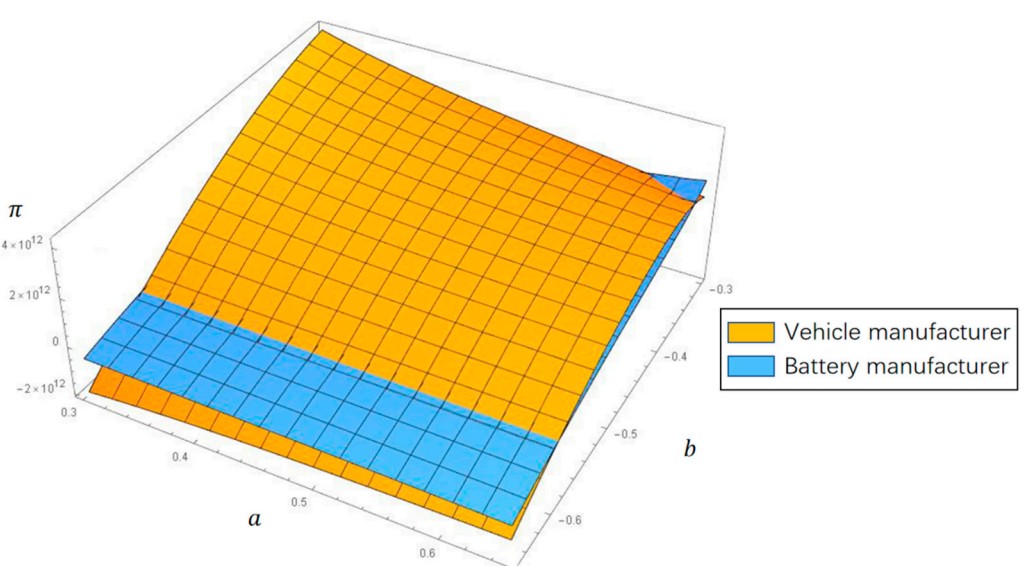

**Figure 20.** The revenue of both enterprises in Scenario V.

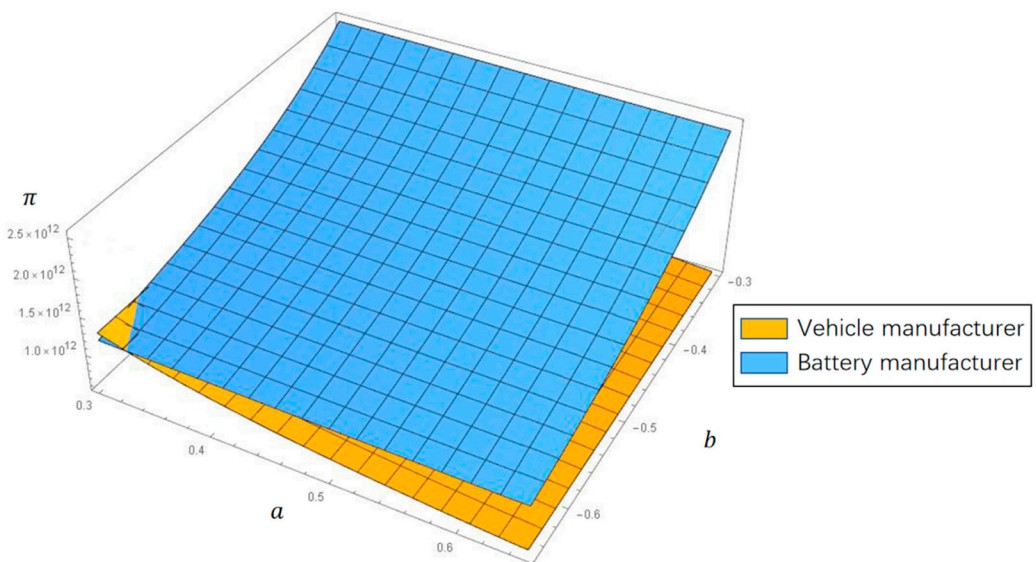

**Figure 21.** The revenue of both enterprises in Scenario B.

The comparison shows that the dominance in the EV supply chain has a more significant impact on the profit of the participants. In all the scenarios, the profits of the leading enterprises with supply chain dominance are significantly higher than those of the following companies without such dominance. The reason for this is that the addition of lease and BSS has transformed the battery from a fixed component into a profitable asset and the power structure of the supply chain has a direct impact on the value distribution of over 40% of the supply chain. At the same time, cascade utilization further amplifies the fluctuation of the value distribution and the realization of the asset.

In Scenario V, compared to Scenario T, the VMs and BMs achieve a significant increase in optimal profits of over 50% under the same power structure. The companies further meet the essential needs of consumers and the emergence of new businesses opens up scope for profit growth up and down the supply chain, offering the possibility of a win–win situation for both.

Comparing Scenario B with Scenario T, it is clear that the BMs can obtain higher profits in the supply chain by breaking down the original structure and providing consumers with BSS across all levels. Therefore, in the rapid development of the EV industry, the BMs

should not be limited to their original role in the supply chain but can actively compete for and extract more profits by providing more services to downstream enterprises or consumers, thus breaking the current market competition pattern.

## 6. Managerial Insights

This research also provides helpful management insights for VMs and BMs.

### 6.1. For VMs Extending the Industry Chain, the Advantages of Battery Swapping Service Outweigh the Disadvantages

For those VMs who tend to provide BSS, what the BSS will bring to them is not only a new revenue growth point and the environmental behavioral benefit of recycling batteries but also an opportunity to extend the industry chain to a whole new level. The company's manufacturing offerings will no longer be limited to EVs and it will become possible to provide finished facilities such as battery swapping stations to the other brands. Through in-depth research and technology development on BSS, once the technology becomes a common standard, the entire industry's technology development direction and path, as well as R&D and production investment, will converge on it, forming a path of technological and industrial development dependence and other technology development solutions will face the danger of being eliminated from the game. It is worth noting that the advantages mentioned above cannot change the fact that BSS is not a business that can be profitable quickly. The long-term lease, the slow realization of cascade utilization, and the lengthy process of drafting standards all dictate that this business will not immediately improve the company's financial statements. Therefore, some enterprises may be forced to use accounting tools such as the one-off inclusion of subscriber subscription fees for the next few years in the current financial statements, which is not a good option.

### 6.2. The Opportunity of Business Transformation and New Profit Growth Is Feasible for BMs by Entering the BSS Market

The invention of the BSS has undoubtedly opened up new growth opportunities for BMs, which are not satisfied with simple battery sales. The price of battery materials has been rising rapidly in recent years due to the sharp rise in EV sales. Although wholesale prices have risen in tandem, it is unrealistic to entirely pass on the raw material increases to downstream companies without breakthroughs in battery manufacturing technology, causing it to be increasingly difficult to increase the profitability of sales. According to a Goldman Sachs research report [53], the prices for battery materials are rising as a result of so-called greenflation. The research outlines their new forecast for a slower pace of decline for automotive battery prices through 2025, which also limit the growth of BM sales profits. By implementing BSS, BMs can skip the downstream VMs in service-oriented manufacture and directly serve end-users, overcoming the current predicament mentioned above. It is worth noting that this transition is premised on sufficient capacity and technological accumulation, which the second-rate BMs cannot afford.

### 6.3. The Escalating Environment and Increased Competition Brought by BSM Forces Second-Rate Manufactures to Cooperate in Depth

The big brands currently implementing the BSM operate independently and have brand advantages and capital scales that second-rate manufacturers cannot match, which is enough to independently bear the high battery swapping stations deployment costs. For those second-rate VMs who cannot dominate the game, the best option is undoubtedly to seek long-term in-depth cooperation with top BMs. The sales of EVs are expected to exhibit continued growth in the coming years. Under such a development momentum, the mainstream automakers joined the automotive industry to seize the market share and battery capacity extruded heavily already [54]. How to ensure battery supply and how to guarantee the user experience are the first issues they should consider, while compromises such as adapting to other brands of swapping stations and batteries can be considered.

What users need in their daily use of the car is not data on paper, but a genuinely excellent energy replenishment experience, which is the key to the growth of the corporate brand.

*6.4. In All Scenarios, Cascade Utilization, as an Important Component and a Valuable Source of Profit, Should Be Given Strong Consideration*

Cascade utilization is a vital part of the closed-loop logic of BSS and leading companies should focus on developing the recycling in either scenario. As the development of renewable energy generation accelerates, energy storage technology will be vital in responding to daytime and seasonal fluctuations in electricity demand, which is a crucial area for cascade utilization. Batteries are currently the most advanced energy storage technology for daytime power generation and, according to a Goldman Sachs study [55], they will outperform other pathways, such as hydrogen, by 30%–50% in overall energy storage efficiency. To achieve China's 2060 carbon neutral target, utility-scale energy storage batteries will exceed 400 GWh, significantly increasing the demand for batteries, especially decommissioned batteries for recycling. If the power load can be predicted, it is expected that the battery swapping station will become a controllable energy storage power station. This will enable the cascade utilization function of the battery to be achieved before decommissioning. Although additional two-way power facility deployments will add to part of the cost, there is also an additional margin for profit for peak-shaving energy storage in the capacity market and the corresponding electricity sales in the frequency market.

**7. Conclusions**

In this paper, we study the pricing strategy of the BSS for EVs. By considering different supply chain power structures and cascade utilization, we build three models of the supply chains under various scenarios and analyze the impact of the quantity of reserve batteries, the ratio of the decommission batteries recycled with cascade utilization and other parameters on the profits of both enterprises.

The conclusions show that: (1) Under all power structures, the development of BSS will significantly increase service providers' profits. The increase in profits of BMs is more significant than that of VMs because of the cost advantages; (2) With the ratio of the decommission batteries recycled with cascade utilization and the revenue of battery cascade utilization increasing, the profits of service providers will be further enhanced; (3) As one of the critical costs of the BSM, reducing the quantity of reserve batteries stocked in the swapping stations is necessary, especially for VMs with higher battery stock costs; (4) As a complementary production to the battery-swapping EV, the pricing strategy of the BSS is related to the interests of all parties in the supply chain. On the premise of averting to affect consumers' willingness to buy, maintaining a reasonable pricing range can achieve win–win results for both sides.

In view of the current market demand and competition pattern, enterprises under different supply chain structures all adopt penetration pricing to promote the BSM. According to the previous research, vehicle manufacturers have more advantages in this situation. In addition, consumers are still extremely sensitive to the pricing of EVs, including the complementary product and specific running costs. The EV market is still in the process of rapid expansion but a lack of profit space. Hence, compared with the second-rate enterprises, leading manufacturers are more capable of increasing early investment to seize the market share. However, due to the huge initial investment required for the battery-swapping EVs, even the leading enterprises are not willing to bear the expense at one time, which means there will be no room for subsequent transformation. How to coordinate the development of rechargeable EVs and battery-swapping EVs will be the focus of enterprises in the next stage.

The contribution of this paper is to study and analyze the Stackelberg decision-making of EV supply chain enterprises considering cascade utilization from a supply chain perspective and considering the differential power structures of BMs and VMs. In addition, this paper provides ideas for improving and promoting the operation of the BSS and the

research results have important guiding significance for the relevant enterprises who have already entered the BSM market or will enter, so as to exert battery maximum value, reduce environment damage, and increase industry sustainability.

This research studies the pricing and operation strategy of the BSM from the perspective of supply chain and can be further extended in several directions to achieve broader insights. Firstly, as an essential way of replenishing energy for EVs, the BSS still cannot completely replace the role of charging at present, so the interaction of both can be considered. Secondly, given the large gap in revenue between the two parties in the model, supply chain coordination with revenue sharing should be the focus of subsequent research. Thirdly, the government subsidy mechanism for environmental behaviors can be taken into account.

**Author Contributions:** Writing, T.W.; Supervision and editing, G.L. All authors have read and agreed to the published version of the manuscript.

**Funding:** This research was funded by [Project of Social Science Foundation of Jiangsu Province], Grant No. [14GLB008].

**Institutional Review Board Statement:** Not applicable.

**Informed Consent Statement:** Not applicable.

**Data Availability Statement:** All data, models, and code generated or used during the study appear in the submitted article.

**Conflicts of Interest:** The authors declare no conflict of interest.

**Appendix A**

To confirm the concavity, we carry out the following calculations to build up the Hessian matrix of $\pi_B^T$ and obtain the first and second order derivatives of $\pi_B^T$ to $h$ and $\omega_b$.

$$H_B^T = \begin{bmatrix} \frac{\partial^2 \pi_B^T}{\partial h^2} & \frac{\partial^2 \pi_B^T}{\partial h \partial \omega_b} \\ \frac{\partial^2 \pi_B^T}{\partial \omega_b \partial h} & \frac{\partial^2 \pi_B^T}{\partial \omega_b{}^2} \end{bmatrix} = \begin{bmatrix} -g_1 & \theta \\ \theta & -2a \end{bmatrix} \tag{A1}$$

The principal minor sequences of the discrimination matrix are $\left|H_B^T\right|_1 = -g_1 < 0$, $\left|H_B^T\right|_2 = 2ag_1 + \theta^2 > 0$, which imply that $\pi_B^T$ is a concave function to $(h, \omega_b)$.

Through setting $\frac{\partial \pi_B^T}{\partial h}$ and $\frac{\partial \pi_B^T}{\partial \omega_b}$ to zero simultaneously, we can obtain the response function of the BM in Scenario T.

$$\frac{\partial \pi_B^T}{\partial h} = \theta(\omega_b - C_b) - g_1 h = 0, \quad \frac{\partial \pi_B^T}{\partial \omega_b} = \phi - aP_v + \theta h + aC_b - a\omega_b = 0 \tag{A2}$$

$$\omega_b{}^* = \frac{g_1 \phi - C_b \theta^2 - ag_1(P_v - C_b)}{2ag_1 - \theta^2}, \quad h^* = \frac{\theta(\phi - aC_b - aP_v)}{2ag_1 - \theta^2} \tag{A3}$$

Then, we substitute the $h^*$ and $\omega_b{}^*$ into Equation (3), which is the objective function of the VM. Additionally, then, the first and second order derivatives of $\pi_V^T$ to $P_v$ can be calculated.

$$\frac{\partial \pi_V^T}{\partial P_v} = \frac{ag_1 \left[\phi + a(C_v - C_b + m - 2P_v - V_a)\right]}{2ag_1 - \theta^2} \tag{A4}$$

$$\frac{\partial^2 \pi_V^T}{\partial P_v{}^2} = \frac{-2ag_1}{2ag_1 - \theta^2} < 0 \tag{A5}$$

It implies the concavity of $\pi_V^T$. Setting $\frac{\partial \pi_V^T}{\partial P_v}$ to zero, the best response function of the VM is:

$$P_v{}^* = \frac{\phi + a(m + C_v - C_b - V_a)}{2a} \tag{A6}$$

Then, we can substitute the $P_v{}^*$ into the equation of the $h^*$ and $\omega_b{}^*$.

$$h^* = \frac{\theta[\phi + a(V_a - m - 3C_v + C_b)]}{4ag_1 - 2\theta^2} \tag{A7}$$

$$\omega_b{}^* = \frac{g_1\phi + ag_1(V_a + C_v + C_b - m) - \theta^2 C_c}{4ag_1 - 2\theta^2} \tag{A8}$$

**Appendix B**

The first and second order derivatives of $\pi_B^V$ to $\omega_b$ can be calculated.

$$\frac{\partial \pi_B^V}{\partial \omega_b} = \lambda(\phi - aP_v + 2bk\omega_b + \theta h) - \lambda bk C_b \tag{A9}$$

$$\frac{\partial^2 \pi_B^V}{\partial \omega_b{}^2} = 2\lambda bk < 0 \tag{A10}$$

It implies the concavity of $\pi_B^V$. Setting $\frac{\partial \pi_B^V}{\partial \omega_b}$ to zero, the best response function of the BM is:

$$\omega_b{}^* = \frac{bk C_b + aP_v - \phi - \theta h}{2bk} \tag{A11}$$

$$\pi_B = (P_s - C_o)Q - \lambda C_b Q + V_a(1 - f)\lambda Q + V_b f \lambda Q - \frac{1}{2}g_3 h^2 \tag{A12}$$

Then, we substitute the $\omega_b{}^*$ in Equation (A12), which is the objective function of the VM. Additionally, then, to confirm the concavity, we carry out the following calculations to build up the Hessian matrix of $\pi_V^V$ and obtain the first and second order derivatives of $\pi_V^V$ to $h$ and $P_v$.

$$H_V^V = \begin{bmatrix} \frac{\partial^2 \pi_V^V}{\partial h^2} & \frac{\partial^2 \pi_V^V}{\partial h \partial P_v} \\ \frac{\partial^2 \pi_V^V}{\partial P_v \partial h} & \frac{\partial^2 \pi_V^V}{\partial P_v{}^2} \end{bmatrix} = \begin{bmatrix} -g_2 - \frac{\theta^2(k-\lambda)}{2bk} & \frac{1}{2}\theta\left(1 + \frac{a(k-\lambda)}{bk}\right) \\ \frac{1}{2}\theta\left(1 + \frac{a(k-\lambda)}{bk}\right) & -a\left(1 + \frac{a(k-\lambda)}{2bk}\right) \end{bmatrix} \tag{A13}$$

The principal minor sequences of the discrimination matrix are $|H_V^V|_1 = -g_2 - \frac{\theta^2(k-\lambda)}{2bk} < 0$ and $|H_V^V|_2 = a\left(g_2 + \frac{\theta^2(k-\lambda)}{2bk}\right) + a^2 g_2 \frac{(k-\lambda)}{2bk} + a^2\theta^2 \frac{(k-\lambda)^2}{4b^2k^2} + \frac{1}{4}\theta^2\left(1 + \frac{a(k-\lambda)}{bk}\right)^2 > 0$, which imply that $\pi_V^V$ is a concave function to $(h, \omega_b)$.

Through setting $\frac{\partial \pi_V^V}{\partial h}$ and $\frac{\partial \pi_V^V}{\partial P_v}$ to zero, we can obtain the response function of the VM in Scenario T. For ease of presentation, we set $C_v + C_o - \lambda(fV_b + V_a - fV_a) = M$.

$$\frac{\partial \pi_V^V}{\partial h} = -g_2 h + \frac{1}{2}\theta\left(\frac{(k-\lambda)(aP_v - \theta h - \phi)}{2bk} + P_v - M\right) = 0 \tag{A14}$$

$$h^* = \frac{bk\theta(bk C_b + \phi - a((\lambda - k)C_b + M)}{4abg_2 k - bk\theta^2 + 2a^2 g_2(k - \lambda)} \tag{A15}$$

$$\frac{\partial \pi_V^V}{\partial P_v} = \frac{a^2 P_v(\lambda - k) + abk(M - 2P_v) + a(k - \lambda)(\theta h + \phi) + bk(\phi + \theta h + bk C_b)}{2bk} = 0 \tag{A16}$$

$$P_v{}^* = \frac{bk[2bC_b g_2 k - \theta^2((\lambda - k)C_b + M) + 2g_2\phi] + 2ag_2(\phi(k - \lambda) + bkM))}{4abg_2 k - bk\theta^2 + 2a^2 g_2(k - \lambda)} \tag{A17}$$

Then, we can substitute the $h^*$ and $P_v{}^*$ into the equation of the $\omega_b{}^*$.

$$\omega_b{}^* = \frac{b^2k^2\left[-bk C_b\theta^2 + ag_2(3bk C_b - \phi) + a^2 g_2((k - \lambda)C_b + M\right]}{4abg_2 k - bk\theta^2 + 2a^2 g_2(k - \lambda)} \tag{A18}$$

$$P_s{}^* = \frac{b^2 k^3 \left[ -bkC_b\theta^2 + ag_2(3bkC_b - \phi) + a^2 g_2((k - \lambda)C_b + M] \right]}{4abg_2 k - bk\theta^2 + 2a^2 g_2(k - \lambda)} \tag{A19}$$

**Appendix C**

According to Equation (A20), the first derivatives of $P_v{}^*$ to $k$ can be calculated as follows:

$$P_v{}^* = \frac{bk\left[2bC_b g_2 k - \theta^2((\lambda - k)C_b + M) + 2g_2\phi\right] + 2ag_2(\phi(k - \lambda) + bkM))}{4abg_2 k - bk\theta^2 + 2a^2 g_2(k - \lambda)} \tag{A20}$$

$$\frac{\partial P_v{}^*}{\partial k} = \frac{b\left(bC_b k^2\left(2bg_2 + \theta^2\right)\left(4ag_2 - \theta^2\right) - 2ag_2\theta^2\lambda\phi - 4a^3 g_2{}^2\lambda M\right)}{\left(4abg_2 k - bk\theta^2 + 2a^2 g_2(k - \lambda)\right)^2} \\ + \frac{2a^2 bg_2\left(2bC_b g_2 k(k - 2\lambda) + C_b\theta^2(k - \lambda)^2 + 2g_2\phi + \theta^2\lambda M\right)}{\left(4abg_2 k - bk\theta^2 + 2a^2 g_2(k - \lambda)\right)^2} \tag{A21}$$

Since $g_2$ should be much greater than $\theta^2$ and $b < 0$, when $k < 2\lambda$, it can be derived that $\frac{\partial P_c{}^*}{\partial k} < 0$. Under vehicle manufacturer dominance, the vehicle's selling price decreases with the increase in the price of the BSS to the wholesale price of batteries when k $< 2\lambda$.

**Appendix D**

According to Equation (A20), the first derivatives of $P_v{}^*$ to $k$ can be calculated as follows:

$$\frac{\partial P_c{}^*}{\partial V_b} = \frac{-2abfg_2 k\lambda + bf\theta^2 k\lambda}{4abg_2 k - bk\theta^2 + 2a^2 g_2(k - \lambda)} \tag{A22}$$

Since $g_2$ should be much greater than $\theta^2$, when $a + 2b < 0$, it can be derived that $\frac{\partial P_c{}^*}{\partial V_b} < 0$. The vehicle selling price will decrease with increasing cascade utilization gains $a + 2b < 0$ when in Scenario V.

**Appendix E**

The first and second order derivatives of $\pi_V^B$ to $P_v$ can be calculated.

$$\frac{\partial \pi_V^B}{\partial P_v} = \phi - aP_v - a(P_v - C_v) + bP_s + \theta h \tag{A23}$$

$$\frac{\partial^2 \pi_V^B}{\partial P_v{}^2} = -2a < 0 \tag{A24}$$

It implies the concavity of $\pi_V^B$. Setting $\frac{\partial \pi_V^B}{\partial P_v}$ to zero, the best response function of the BM is:

$$P_v{}^* = \frac{\phi + aC_v + bP_s + \theta h}{2a} \tag{A25}$$

$$\pi_B = (P_s - C_o)Q - \lambda C_b Q + V_a(1 - f)\lambda Q + V_b f\lambda Q - \frac{1}{2}g_2 h^2 \tag{A26}$$

Then, we substitute the $P_v{}^*$ in Equation (A26), which is the objective function of the BM. Additionally, then, to confirm the concavity, we carry out the following calculations to build up the Hessian matrix of $\pi_B^B$ and obtain the first and second order derivatives of $\pi_B^B$ to h and $\omega_b$.

$$H_B^B = \begin{bmatrix} \frac{\partial^2 \pi_B^B}{\partial h^2} & \frac{\partial^2 \pi_B^B}{\partial h \partial P_s} \\ \frac{\partial^2 \pi_B^B}{\partial P_s \partial h} & \frac{\partial^2 \pi_B^B}{\partial P_s{}^2} \end{bmatrix} = \begin{bmatrix} -g_3 & \frac{1}{2}\theta \\ \frac{1}{2}\theta & b \end{bmatrix} \tag{A27}$$

The principal minor sequences of the discrimination matrix are $\left| H_B^B \right|_1 = -g_3 < 0$, $\left| H_B^B \right|_2 = -bg_3 + \frac{1}{4}\theta^2 > 0$, which imply that $\pi_B^B$ is a concave function to $(h, \omega_b)$.



Through setting $\frac{\partial \pi_B^B}{\partial h}$ and $\frac{\partial \pi_B^B}{\partial \omega_b}$ to zero, we can obtain the response function of the BM in Scenario T.

$$\frac{\partial \pi_B^B}{\partial h} = -g_3 h - \frac{1}{2}\theta(C_o - P_s + (C_b + (f-1)V_a - fV_b)\lambda) = 0 \tag{A28}$$

$$h^* = \theta \frac{aC_v + b(V_a(1-f)\lambda + V_b f\lambda - C_o - \lambda C_b) - \phi}{\theta^2 + 4bg_3} \tag{A29}$$

$$\frac{\partial \pi_B^B}{\partial P_s} = \frac{\phi - aC_v + \theta h - b(C_o - 2P_s + (C_b + (1-f)V_a - fV_b)\lambda)}{2} = 0 \tag{A30}$$

$$P_s{}^* = \frac{g_3(2aC_v - 2\phi) - (\theta^2 + 2bg_3)(V_a(1-f)\lambda + V_b f\lambda - C_o - \lambda C_b)}{\theta^2 + 4bg_3} \tag{A31}$$

Then, we can substitute the $h^*$ and $P_s{}^*$ into the equation of the $P_v{}^*$.

$$P_v{}^* = \frac{4bg_3\phi + a(3\theta^2 + 10bg_3)C_v - 4b^2 g_3(V_a(1-f)\lambda + V_b f\lambda - C_o - \lambda C_b)}{4a(\theta^2 + 4bg_3)} \tag{A32}$$

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
