# Peer review of "Long-Term Leases vs. One-Off Purchases: Game Analysis on Battery Swapping Mode Considering Cascade Utilization and Power Structure"

_sustainability, doi:10.3390/su142416957_

Round 1

Reviewer 1 Report

Title: Long-term Leases VS One-off Purchases: Game Analysis on Battery Swapping Mode Considering Cascade Utilization and Power Structure

Journal: Sustainability

Manuscript ID:

Comment 1: An abstract is not clear. Authors are advised to explain the methodology and key results in the abstract.

Comment 2: authors are advised to include the methodology diagram for a better understanding of carried research work to the reader. 

Comment 3: Literature on the impact of the current supply chain system on environmental issues and customer satisfaction in vehicle and battery manufacturers should be added in the literature review section. 

Comment 4: Numerical comparison of results for the selected methodology in the results and discussion section is missing in the text.

Comment 5: conclusion is not clearly stated for techniques used for supply chain enterprises and authors are advised to include the future prospect of conducted research.

Comment 6: In general, the section of Introduction is well organized, but lacks sufficient literature review. The introduction part should be improved further to review the relevant research and highlights the advantages of your work. 

    https://doi.org/10.1007/s40032-022-00852-6

    https://doi.org/10.1177/09544070221080349

Author Response

Thank you for your profound and kindly suggestions. All your suggestions are very significant, and they are of great guiding significance to my thesis writing and scientific research!

Reviewer 2 Report

sustainability-1993151-peer-review-v1

Long-term Leases VS One-off Purchases: Game Analysis on Battery Swapping Mode Considering Cascade Utilization and Power Structure.

This paper is very nice, strong, and has mathematical real-world applications; I see many positive aspects in this work and would like to see it published. This paper will be of value and interest to a significant portion of potential readers of the journal. They have been taken the advantage of “Decision-making and coordination of green closed-loop supply chain with fairness concern (Ma et al. [36])”. The distinguished features of these vehicle manufacturers (VMs) and battery manufacturers (BMs) are investigated as well.

In view of this, I recommend to accept this manuscript as it is.

Thank you for giving me this opportunity.

Regards,

Author Response

Thank you for choosing to accept this article.

Reviewer 3 Report

In the study, game analysis on battery swapping  model considering cascade utilization and power structure is presented. The work is well presented and explained. One minor comment need to address before the final acceptance:

1. Improve the quality of the figures 4 to 13. Text should be clear to read. 

Author Response

Thank you for your suggestion about the quality of the figures. Words fail me when i want to express my sincere sorry for the unclear text.

Reviewer 4 Report

In my opinion, the manuscript needs to have made some revisions.
In particular, the reviewer suggests to:
1) reinforce the introductory section, by enlarging the considerations derived by studying the literature and and highlighting the conclusions in individual studies;
2) give a launch on the transition to hydrogen-electric hybrid vehicles, where the battery will still play a crucial role, albeit in a spot form - in this regards the authors should include, even small, an inner paragraph or section. Here some suggestions:
https://doi.org/10.1016/j.procs.2022.01.326

https://doi.org/10.3390/wevj13090172

3) include a scheme/block/flow diagram to well present the work the readers are going to find in the article body;

4) put a little more effort into the conclusions section, and resubmit/represent or sinthetize the important numbers that were inferred from the simulation results.

Author Response

Thank you for your profound and kindly suggestions. All your suggestions are very significant, and they are of great guiding significance to my thesis writing and scientific research! Please see the attachment.

Round 2

Reviewer 1 Report

Accept